# Metoclopramide and Propofol to Prevent Nausea and Vomiting during Cesarean Section under Spinal Anesthesia: A Randomized, Placebo-Controlled, Double-Blind Trial

**DOI:** 10.3390/jcm11010110

**Published:** 2021-12-26

**Authors:** Zhirajr Mokini, Valentina Genocchio, Patrice Forget, Flavia Petrini

**Affiliations:** 1Independent Researcher, European Society of Anaesthesiology and Intensive Care Mentorship Programme, B-1000 Brussels, Belgium; 2Ospedale San Maurizio di Bolzano, Via Lorenz Böhler 5, 39100 Bolzano, Italy; valentina.genocchio@sabes.it; 3Epidemiology Group, Institute of Applied Health Sciences, School of Medicine, Medical Sciences and Nutrition, University of Aberdeen, Aberdeen AB25 2ZD, UK; patrice.forget@abdn.ac.uk; 4Department of Anaesthesia, NHS Grampian, Aberdeen AB25 2ZN, UK; 5Department of Emergency, SS.ma Annunziata University Hospital, Via dei Vestini, 66100 Chieti, Italy; flavia.petrini@unich.it

**Keywords:** propofol, metoclopramide, spinal anesthesia, nausea, vomiting, retching

## Abstract

Background: Intra-operative nausea, vomiting and retching (NVR) are frequently associated with subarachnoid anesthesia (SA) in women undergoing cesarean section (CS). In this study performed in women undergoing CS under SA with a risk factor control strategy, we compared saline (placebo), propofol, metoclopramide and both drugs to prevent NVR. Methods: We recorded NVR events in 110 women undergoing CS who were randomized after umbilical cord clamping to receive saline (S; *n* = 27), metoclopramide 10 mg (M; *n* = 28), propofol 1 mg/kg/h (P; *n* = 27) or both drugs (PM; *n* = 28). Results: The proportion of women with intra-operative nausea was: S: 17/27 (63%); P: 15/27 (56%); M: 13/28 (46%); PM: 6/28 (21%) (*p* = 0.012, Cramér’s V = 0.31 (large effect). The proportion of women with intra-operative vomiting/retching was: S: 9/27 (33%); M: 7/27 (25%); P: 3/28 (11%); PM 2/28 (7%) (*p* = 0.049, Cramér’s V = 0.26 (medium effect). Post-hoc multiple comparisons revealed a significant reduction in NVR episodes and NRS scores between the PM group and control. Sedation scores did not differ among groups. Conclusion: In women undergoing CS under SA with a risk factor control strategy, combined propofol and metoclopramide reduce nausea and vomiting.

## 1. Introduction

Intra-operative nausea and vomiting/retching (NVR) may be experienced by 20% to 80% of women undergoing cesarean section (CS) with subarachnoid anesthesia (SA) in the absence of antiemetic prophylaxis [1].

Recommendations for reduction of the incidence of NVR during CS under SA include administration of prophylactic antiemetics including sedative serotonin 5-HT3 antagonists and dopamine receptor antagonists [1,2].

Further recommended interventions are aimed at the prevention of predisposing factors for NVR. Interventions to prevent reduction of preload and afterload as a consequence of aorto-caval compression, hypotension and bradycardia from SA include fluid loading before anesthesia, vasoconstriction, left uterine displacement, Trendelenburg position and anticholinergic antagonism [1,3,4,5]. Slow positioning movements help to prevent vestibular activation [6]. Administration of opioids help to prevent visceral pain from surgical stimulation of abdominal organs, visceral traction or exteriorization of the uterus [7]. Slow administration of uterotonic agents is also recommended [8,9]. Interventions to reduce the risk of acidic aspiration may also affect nausea and vomiting [10].

In this study, we compared placebo saline, propofol (a serotonin 5-HT3 receptor antagonist), metoclopramide (a dopamine antagonist), or their combination to prevent NVR in women undergoing CS under SA.

## 2. Materials and Methods

Following approval from the Ethics Committee of the University of Chieti (No. 1632/08) and written informed consent, we enrolled 112 women, ASA I–II, undergoing elective cesarean section, in a prospective, placebo-controlled, double-blind trial. We conducted the study according to the Helsinki declaration and registered it at www.clinicaltrials.gov (NCT: NCT01781377, accessed on 25.12.2021).

We excluded women in the case of emergency CS with fetal and/or maternal distress, a history of previous complicated pregnancy, contraindications to SA, fetal age < 36 or > 41 weeks, predicted fetal weight < 2.5 kg, BMI > 35, hemoglobin <10 mg/dL, previous major abdominal surgery, a history of smoking or drug addiction during pregnancy, any allergy to study drugs or consumption of antiemetic drugs.

Before surgery, a research assistant confirmed eligibility and provided a consecutively numbered, sealed envelope containing the allocation group to the anesthesiologist who administered the appropriate interventions. Both women and the researcher that collected the data were blinded.

Women followed the same preoperative fasting protocol as typically would be adhered to. Leg compression devices were placed. Premedication included IV ranitidine 50 mg I.V. for prevention of acidic aspiration and IM atropine 0.5 mg I.M. for anticholinergic antagonism [1]. Before anesthesia, we administered 15 mL/kg of Lactated Ringer’s and 500 mL of 6% Hydroxyethyl starch (HES) to all women for the prevention of hypotension after SA [1,4,5]. We used slow movements during patient positioning to prevent vestibular activation and administered 3 L/min of oxygen with nasal cannulas during surgery [11,12].

We performed SA at levels L 2–3, 3–4 or 4–5 using a 22 G, Whitacre needle. We used hyperbaric 5 mg/mL bupivacaine according to weight-height charts and assessed spinal block height by pinprick. After SA we placed a wedge under the right hip (20°–30°) to produce left uterine displacement to allow for maximal caval relief and placed women in the Trendelenburg position (15°–20°) until beginning of surgery [3]. During surgery, we administered IV atropine 0.5 mg if the heart rate was <60 bpm and boluses of ephedrine 5 mg to maintain systolic blood pressure at 90–100% of baseline [13].

After delivery, we allocated women according to a randomly generated computer sequence to one of four interventions (N = 28 in each group): Group S received saline infusion, group M received IV metoclopramide 10 mg in 100 mL saline, group P received an IV propofol bolus of 10 mg followed by an infusion of 1 mg/kg/h (based on pre-pregnancy weight), group PM; received both propofol and metoclopramide. Postoperative rescue antihemetics included metoclopramide 10 mg and/or ondansetron 4 mg.

The 24 h level of hypnotic doses of propofol used for general anesthesia are low in human milk thus, are safe and do not necessitate cessation of lactation [14].

The researcher that collected the data was blinded to the interventions. We withdrew women from the study in the case of protocol deviation or if general anesthesia was necessary. Excluded women received the same in-hospital care and postoperative follow-up for safety analysis.

### 2.1. Measures and Outcomes

The primary outcome was the incidence of nausea as measured from delivery to the end of surgery. We considered nausea being a subjectively unpleasant sensation associated with awareness of the urge to vomit. We considered vomiting as characterized by labored, spasmodic and rhythmic contraction of the respiratory and abdominal muscles aimed at the forceful oral expulsion of gastric contents.

We instructed women preoperatively to use a numerical ranking scale (NRS, 0 = no symptoms, 10 = worst symptoms ever experienced) for NVR which, in this context, was considered easy to use intraoperatively, and possibly more accurately capturing the subjective feeling of nausea [15]. We recorded episodes of NVR, NRS scores and rescue anti-emetic use before SA, after SA, after umbilical cord clamping, during abdominal exploration and at the end of surgery.

We collected the following intra-operative data before anesthesia, after anesthesia, after umbilical cord clamping, during abdominal exploration and at the end of surgery: oxygen saturation, heart rate, blood pressure, sedation scores (0 = awake, 1 = sedated, 2 = arousable, 3 = deep sedation) and headache (Y/N). At the end of surgery we recorded Y/N for the following data: blood loss (mL), need for uterine massage, uterine exteriorization, abdominal exploration, visceral manipulation, peritoneal traction, uterotonic supplementation, epigastric discomfort, vertigo, shoulder pain and any other adverse events. We administered the maternal satisfaction questionnaire to women 2 h after surgery [16]. We translated it in Italian since to our knowledge there is no validated Italian questionnaire for women undergoing CS under SA (Appendix A). We also recorded Apgar scores and fetal gas analyses. We assessed development of aspiration pneumonia until discharge.

### 2.2. Statistical Analysis

We previously performed a pilot observational study in our hospital where the intraoperative NVR was 65% without prophylaxis. Using these data, we calculated that 25 women per group would provide a power of 0.9 and a significance level of 0.05 for a reduction of 0.45 in the incidence of nausea. The sample size was adjusted to correct for continuity to 28 women per group. There was a pre hoc decision to analyze all data according to an intention-to-treat approach.

We used the Shapiro–Wilk test to verify the normality of distribution of continuous variables. We compared normally distributed data with univariate ANOVA. We analyzed categorical data with non-exact 2-tailed chi test (χ^2^) or 2-tailed Fisher’s exact test.

We calculated the effect size for proportions with Cramér’s V (φc). Cramér’s V is commonly used to describe the effect size between categorical variables for a contingency table larger than 2 × 2. A value for Cramer’s V between 0.06 and 0.17 indicates a small effect, a value between 0.17 and 0.29 is a medium effect, and a value greater than 0.29 is considered a large effect [17]. We compared non-parametric data with ANOVA on ranks. We planned to perform post hoc intergroup comparisons (No. 3) with Dunnett’s one-tailed test (D_i_) [18]. With a sample size of 28 patients per group and a significance level of 0.05 the Dunnett’s critical value for 4 groups is 2.47. We compared the Dunnett’s critical value with differences in group means. We adjusted the *p*-value to control for family-wise error rate according to the Bonferroni method (0.05/3 comparisons against control group = 0.017). Normally distributed data are presented as a mean (SD). Data that did not fit a normal distribution are presented as a median (range).

We considered *p* < 0.05 statistically significant. We computed statistical comparisons using the Statistical Package for the Social Science (SPSS for Windows, release 23.0; Chicago, IL, USA).

## 3. Results

We assessed 158 women for eligibility, excluded 46 women and enrolled 112 women in the study (Table 1).

We excluded one woman from group S and group P because of protocol deviations. We found hypothyroidism and gastro-esophageal reflux disease (GERD) being the most frequent concurrent diseases without statistical difference among groups. We also found no differences in the proportion of women that were taking anti-acids for GERD. One patient in each of group S, P and PM had previously taken anti-emetics for NVR during pregnancy. We did not find statistical differences among the following: ASA scores, duration of surgery or duration of the post-delivery interval (Table 2). No patient presented a spinal block above T5. Intra-operative and fetal characteristics are also shown in Table 2. During CS, a uterine myoma was removed in one woman in group P and PM.

Primary analysis showed differences among groups in the incidence of nausea (*p* = 0.008), vomiting/retching (*p* = 0.047) and overall NVR episodes (*p* = 0.006) from delivery to the end of surgery (Table 3). Secondary post hoc intergroup comparisons revealed significant differences in median episodes of nausea between the PM group and group S. However, we found no differences between group S, M and P (Table 3). The difference was significant for vomiting/retching episodes among groups PM and S and for cumulative NVR episodes between group PM and groups S and M (Table 3).

There was a significant reduction in the proportions of women with intra-operative nausea or vomiting with a large to medium effect size between PM vs. S group as shown in Table 4.

We found also significant differences at post hoc testing for maximum intraoperative NRS scores for NV between PM and S groups and for V between P and S groups as shown in Table 5.

Sedation scores did not differ among groups, with the highest sedation score from delivery to the end of surgery was two reported by one woman in group P and three women in group PM (Table 6).

Significantly less patients having received propofol presented headache during the study (Table 6). We found no differences in the proportion of patients with a systolic blood pressure below 90 mmHg and a diastolic blood pressure below 50 mmHg (Appendix A). One woman (4%) in group P experienced an episode of bradycardia. We found no differences in the proportion of patients reporting vertigo or epigastric discomfort. Nine patients in group PM (32%) reported shoulder pain compared to 5 (17%) in group M, 1 in group P (4%) and 0 in group S (*p* = 0.006, χ^2^ test, Cramér’s V = 0.37). One women in group M (4%), P (4%), and PM (4%) had an erythematous reaction over the face and trunk at the time of oxytocin and methylergonovine administration. No patients developed extrapyramidal side effects or aspiration pneumonia. We did not find significant differences among groups in mean Maternal Satisfaction Questionnaire scores reported 2 h post-operatively (*p* = 0.1, Appendix A).

Fetal extraction was difficult in one woman in group S (4%). We did not record any episodes of fetal heart rate below 100, neonatal asphyxia, neurological disorders or trauma during CS. One newborn in group M (4%) had transitory apnea. All newborns had adequate heart rate at birth and there were no differences in the Apgar scores or fetal blood gas values amongst the groups.

## 4. Discussion

This study shows that in women where a risk control strategy is administered, the preventive combination propofol and metoclopramide reduce intraoperative NVR during CS under SA.

While there is good evidence that single agents are effective in preventing nausea, there is limited data comparing different combinations of treatment [1]. In this trial, however, single agents showed no differences compared to control except for Propofol alone that reduced the proportion of women with vomiting (RRR 0.66 (95% CI: −0.09–0.8)) and the NRS scores of vomiting (D_i_ = 0.011).

On the other side, we found that PM administration significantly reduced nausea episodes and NV scores compared to control group (Table 3). Moreover, the proportion of women reporting nausea or vomiting/retching was significantly lower (with a “large” effect size) for the PM group (Table 3).

The difference in vomiting/retching episodes was not significant between groups although between the PM group and the saline (S) group it reached a D_i *p*_ = 0.017 (Table 3). Since vomiting/retching is a less frequent symptom compared to nausea alone, perhaps a greater sample size could have been necessary to see a greater difference. On the other hand, while nausea is a subjective feeling, vomiting/retching is a more objective outcome. For this reason, since in this trial propofol alone or in combination did not reduce episodes of vomiting/retching but reduced NRS scores, it is possible that this effect is to be attributed to the sedative effect of propofol instead of an additive antiemetic effect.

Prevention and treatment of NVR during CS includes pharmacologic interventions and measures against predisposing factors. Emesis is mediated by serotoninergic 5-HT3 receptors, dopaminergic DA2 receptors, muscarinic M1 receptors, opioid receptors, enkefalin and histaminic H1 receptors. Propofol is a sedative with 5-HT3 antagonism activity in the area postrema, chemoreceptor trigger zone (CHRTZ) and gastrointestinal tract [19,20,21,22,23,24]. The antiemetic effect of propofol in the area postrema is elicited through activation of GABA-A receptors that inhibit serotoninergic activity [21,22,23]. Propofol has demonstrated a reduction in intra-operative nausea and vomiting during CS under SA for cesarean section [1]. In this trial we did not find any differences in sedation among patients, which indicates that administration of subhypnotic propofol is safe. Metoclopramide is an effective antiemetic in obstetric patients undergoing SA that acts by antagonizing dopamine receptors in the (CHRTZ) and the gastrointestinal tract [25]. To our knowledge, no study has combined propofol and metoclopramide for the prevention of NVR during CS under SA.

In this study all women received interventions against risk factors in order to reduce NVR incidence at baseline. In the general population, the presence of one or more risk factors is associated with a progressively increased incidence of NVR [26] while risk factors in pregnant women undergoing SA are less clear [27]. In non-obstetric women, spinal anesthesia, subarachnoid vasoconstrictors, heart rate <60 bpm, a block height > T5, a systolic blood pressure <80 mmHg, opioid use, and a history of motion sickness are recognized risk factors for intra-operative NVR. Pregnancy is associated with a physiologically hyperemetic state in which increased progesterone and estrogen levels alter gastrointestinal motility and reduce the threshold of nausea and vomiting centers for NVR [20,28]. The rise in intraabdominal pressure following increased uterine volume further increases emetic stimuli [28].

SA depresses the sympathetic system and leads to a prevalence of the parasympathetic system that causes vasodilation, bradycardia and hypotension. Consequently, there is reduced blood flow to the NVR centers in the brainstem and increased gastrointestinal hyperactivity mediated by the Vth cranial nerve [23]. The incidence of NVR is reduced if blood pressure is maintained at 90–100% of baseline during CS under SA [13]. Fluid loading alone before anesthesia may be not sufficient to prevent hypotension and NVR, whereas the use of vasoconstrictors effectively reduces the incidence of NVR [5]. Moreover, the preventive treatment of hypotension is more effective than symptomatic treatment [5].

Surgical manipulation and consequent visceral pain, blood loss, uterotonic agents and antibiotics increase the risk of intra-operative NVR [23,28]. Slow administration of uterotonics and other measures such as slow positioning movements may also help reduce vestibular activation and were considered in this study.

In this trial, while combined treatment was effective against NVR, the effect was less evident with single treatment groups. Preventive measures against risk factors could have reduced baseline NVR risk and covered the expected differences in NVR episodes and their severity between the placebo saline group (S) and single pharmacologic prophylaxis groups (M and P), thus allowing us to see significant differences only for the combined prophylaxis group (PM).

This trial includes some limitations. The study is underpowered to detect small differences among active groups. There was also a difference on uterine exteriorization and headache among groups which may influence intraoperative NVR (Table 2 and 6). Serotonin 5-HT3 receptor antagonists were not included that should be evaluated in the context of multiple intervention strategies for the prevention of intra-operative risk factors of NVR. Another limitation is the fact that after having translated the Maternal Satisfaction Questionnaire into Italian, we did not retranslate it again by a native speaker from Italian into English to retain content validity and compare the original English version to the one obtained after double translation. Finally, a center-specific effect cannot be excluded due to the design, and may have had an influence, for example, on the choice of drugs such as local anesthetics or vasoactive agents which, in turn, may have influenced the results. However, the strengths linked the design (blinded and, more importantly, randomized) may have, at least partially, neutralized this risk.

In conclusion, this study found that combined intra-operative propofol and metoclopramide reduced total episodes of intraoperative nausea, reduced the severity of nausea and vomiting, and reduced the proportion of patients presenting with NVR when compared to placebo in women undergoing CS under SA. Single agents propofol and metoclopramide did not reduce total episodes of intraoperative nausea and vomiting compared to control. Satisfaction scores and postoperative rescue antiemetic use did not differ among patients. These findings show that combined prophylaxis may have a beneficial role in the prevention of NVR during CS with SA and that further studies with other drugs should be performed in this field.

## Figures and Tables

**Table 1 jcm-11-00110-t001:** CONSORT flow diagram.

Enrollment	Assessed for Eligibility (*n* = 158)	Excluded (*n* = 46) Not Meeting Inclusion Criteria (*n* = 40) Declined to Participate (*n* = 6)		
	Randomized (*n* = 112)			
Allocation	S—Allocated to intervention (*n* = 28) Received intervention (*n* = 28)	M—Allocated to intervention (*n* = 28) Received Intervention (*n* = 28)	P—Allocated to intervention (*n* = 28) Received intervention (*n* = 28)	PM—Allocated to intervention (*n* = 28) Received Intervention (*n* = 28)
Follow-up	Lost to follow-up or discontinued intervention (*n* = 0)	Lost to follow-up or discontinued intervention (*n* = 0)	Lost to follow-up or discontinued intervention (*n* = 0)	Lost to follow-up or discontinued intervention (*n* = 0)
Analysis	Analyzed (*n* = 27) Excluded from analysis because converted to general anesthesia (*n* = 1)	Analyzed (*n* = 28)	Analyzed (*n* = 27) Excluded from analysis because converted to general anesthesia (*n* = 1)	Analyzed (*n* = 28)

**Table 2 jcm-11-00110-t002:** Characteristics of women before and during surgery.

Preoperative Data	S (27)	M (28)	P (27)	PM (28)	
Age, year	34 (3)	34 (3.5)	34 (5)	34 (3)	
Weight before pregnancy, kg	63 (8.5)	66 (12.5)	60 (15)	63 (10)	
Actual weight, kg	78 (7)	76 (12)	75 (11)	76 (11)	
Height, cm	166 (6)	163 (7)	163 (5.5)	165 (5.5)	
Actual BMI, kg/m^2^	28 (3)	28,5 (4)	28 (4)	28 (4)	
Parity, *n*	1 (0–2)	1 (0–5)	1 (0–4)	1 (0–3)	
Age of first pregnancy, year	30 (4)	31 (3.5)	28 (5)	29.5 (3)	
Previous CS, yes/no	1 (0–2)	1 (0–2)	1 (0–4)	1 (1–1)	
NVR during previous CS,	22%	43%	22%	39%	
NVR during actual pregnancy,	74%	53.5%	52%	57%	
Month of beginning of NVR,	I (I–III)	I (I–III)	I (I–V)	I (I–II)	
Month of end of NVR,	III (III–IX)	III (II–IX)	III (III–VIII)	III (III–IX)	
Hypotension,	56%	43%	38%	61%	
Motion sickness,	37%	18%	30%	18%	
Anesthesia and intra-operative data					*p-*value
Bupivacaine dose, mg	12 (1)	12 (1)	12 (1)	12 (1)	0.2
Ephedrine dose, mg	21 (12)	18 (12)	23 (18.5)	15 (15)	0.14
Spinal level,	L2 11% L3 74% L4 15%	L2 0% L3 54% L4 46%	L2 15% L3 63% L4 22%	L2 4% L3 68% L4 28%	
Uterine exteriorization	25 (92%)	17 (60%)	13 (48%)	22 (78%)	0.002 * χ^2^
Intraabdominal manipulation	26 (96%)	26 (93%)	24 (88%)	28	0.35
Peritoneal traction	2 (8%)	4 (14%)	1 (4%)	1 (3%)	0.37
Uterine massage	25 (92%)	20 (71%)	20 (75%)	22 (78%)	0.2
Supplemental ergonovine	1 (4%)	1 (3%)	2 (8%)	2 (6%)	0.8
Supplemental ergonovine, fl	0 (0–2)	0 (0–2)	0 (0–1)	0 (0–1)	1
Supplemental oxytocine	8 (30%)	11 (39%)	5	7 (25%)	0.37
Supplemental oxytocine, fl	0 (0–2)	0 (0–2)	0 (0–1)	0 (0–2)	0.3
Blood loss, ml	325 (104)	350 (166)	370 (171)	305 (171)	0.4
Fetal data					
Apgar 1	9 (6–9)	8 (6–10)	8 (7–9)	8 (6–9)	0.8
Apgar 2	9 (8–10)	9 (8–10)	9 (8–10)	9 (8–10)	0.9
pH	7.34 (0.5)	7.34 (0.4)	7.33 (0.6)	7.36 (0.3)	0.2
Lactate	2 (1)	2 (0.6)	3.5 (7)	1.6 (0.5)	0.38
Blood glucose	70 (9)	65 (15)	68 (8)	71 (16)	0.37
Fetal Hemoglobin	15 (2)	15 (2)	16 (2)	15 (2)	0.75
Base excess	−3.5 (2.6)	−3.1 (2)	−3.2 (2.3)	−1.8 (2)	0.053
pO_2_	27 (6)	30 (10)	29 (9)	28 (4)	0.8
pCO_2_	40 (4.5)	40 (7)	40 (10)	39 (6.5)	0.9

Data are shown as mean (SD), median (range) or proportions of women (%). NVR = nausea, vomiting and retching, CS = cesarean section. Parametric data are analyzed with ANOVA. Proportions are analyzed with 2-tailed χ^2^ = chi/Fisher tests. * Difference in descriptive data is significant at 0.05.

**Table 3 jcm-11-00110-t003:** Mean episodes of NVR after delivery.

Variable	S (27)	M (28)	P (27)	PM (28)	*p*-Value	D_i_ Post Hoc Test (<S)
N, *n*	0.8 (0.7)	0.6 (0.8)	0.6 (0.7)	0.2 (0.7)	0.008 *	M vs. S = 0.3 P vs. S = 0.4 PM vs. S = 0.002 *
V/R, *n*	0.4 (0.6)	0.3 (0.6)	0.15 (0.45)	0.1 (0.2)	0.047 *	M vs. S = 0.4 P vs. S = 0.05 PM vs. S = 0.017
NVR, *n*	1.2 (1.1)	1 (1.2)	0.8 (0.9)	0.2 (0.5)	0.006 *	M vs. S = 0.2 P vs. S = 0.2 PM vs. S < 0.001 *
Rescue anti-emetic, *n*	4 (15)	3 (7)	4 (15)	1 (3.5)	0.4	

N = nausea, V/R = vomiting/retching. We summarize data as mean ± sd and number of patients (%). Non-parametric data are analyzed with ANOVA on ranks. * Difference in descriptive data is significant at 0.05. Significant post hoc differences with Dunnett’s test (Di) at 0.017 are shown.

**Table 4 jcm-11-00110-t004:** Proportions of women presenting with nausea or vomiting.

Variable	S (27)	M (28)	P (27)	PM (28)	*p*-Value	Effect Size φ_c_	Relative Risk Reduction
N, *%*	17/27 (63%)	M: 13/28 (46%)	P: 15/27 (56%)	PM: 6/28 (21%)	0.012 *	0.31	PM vs. S = 0.66 (95% CI: 0.2–0.8) M vs. S = 0.26 (95% CI: −0.2–0.5) P vs. S = 0.11 (95% CI: −0.3–0.4)
V/R, *%*	S: 9/27 (33%)	M: 7/27 (25%)	P: 3/28 (11%)	PM 2/28 (7%)	0.049 *	0.26	PM vs. S = 0.78 (95% CI: 0.09–0.9) P vs. S = 0.66 (95% CI: −0.09–0.8) M vs. S = 0.22 (95% CI: −0.7–0.6)

N = nausea, V/R = vomiting/retching. We present data as number of patients and (%). Proportions are analyzed with 2-tailed χ^2^/Fisher tests. * Difference is significant at 0.05. Cramér’s V = φ_c_ (0.17 to 0.29 medium effect, >0.29 = large effect).

**Table 5 jcm-11-00110-t005:** Intraoperative NRS scores of NVR.

Variable	S (27)	M (28)	P (27)	PM (28)	*p*-Value	D_i_ Post Hoc Test (<S)
N, *NRS*	3.7 (3.5)	3.9 (4.5)	3.4 (3.8)	1.1 (2.4)	0.017 *	PM vs. S = 0.008 *
V/R, *NRS*	3.3 (4.8)	2.1 (3.9)	0.7 (2.2)	0.6 (2.3)	0.015 *	PM vs. S = 0.008 * P vs. S = 0.011 *
NVR, *NRS*	5 (4.3)	4.1 (4.7)	3.7 (4.1)	1.6 (3.2)	0.023 *	PM vs. S = 0.004 *

N = nausea, V/R = vomiting/retching, NRS = numerical ranking scale. We present data summarized as mean ± SD analyzed with ANOVA. * Difference in descriptive data is significant at 0.05. Significant post hoc differences with Dunnett’s test = D_i_ at 0.017 are shown.

**Table 6 jcm-11-00110-t006:** Sedation, headache and Maternal Satisfaction Questionnaire scores.

Variable	S (27)	M (28)	P (27)	PM (28)	*p*-Value	Effect Size φ_c_
Sedation from delivery to the end of surgery, %	6 (22)	6 (21)	10 (37)	14 (50)	0.07	
Headache from delivery to the end of surgery, %	14 (55)	12 (43)	8 (30)	3 (10)	0.004 *	0.34

Data are summarized as mean ± sd and number of patients (%). Non-parametric data are analyzed with ANOVA on ranks. Proportions are analyzed with 2-tailed χ^2^/Fisher tests. * Difference in descriptive data is significant at 0.05. Significant post hoc differences are shown: Cramér’s V = φ_c_ (0.17 to 0.29 medium effect, >0.29 = large effect).

## Data Availability

All the data are included in the manuscript.

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
