# Peer review of "Metoclopramide and Propofol to Prevent Nausea and Vomiting during Cesarean Section under Spinal Anesthesia: A Randomized, Placebo-Controlled, Double-Blind Trial"

_jcm, 2021, doi:10.3390/jcm11010110_

Round 1

Reviewer 1 Report

The authors performed a prospective, placebo-controlled, double blind trial to investigate the effects of propofol, metoclopramide or both to prevent nausea and vomiting during cesarian section under spinal anesthesia.

The outcome that propofol as well as reduce nausea and vomiting in this setting is already known. (Ref 1, 2,3). Thus part of this study is only confirmatory. However the grade of evidence is in the metanalysis (1) is low and the current manuscript may add to the evidence.

The result that the combination of propofol and metoclopramide is more effective in preventing nausea than either substance alone is a novel finding.

However, the combination did not prevent vomiting more efficiently than either substance alone. As the authors themselves state in the manuscript, the study probably was underpowered to detect the small differences.

Specific comments:

1

Was the thoracic level of SA assessed? Could differences in anesthesia >Th5 account for the differences observed?

2

It is not clear to me which post test refers to which variable: 3 variables, 6 post tests, partially vs P partially vs M. The lines of the row post test do not correspond to the lines of the row variable

It is not clear what “Di” means

3

Fig 1 the graph simulates a timeline. Instead 3 groups are compared. Use columns instead of lines.

4

Discussion:

The Part from “Spinal anesthesia” page 7 line 240 to p8 line 257 is of general interest but does not discuss the results. Please tie the literature review to your findings.

5

The authors should more clearly point out the novel finding and also clearly state what part of the research is confirmatory.

It should also be pointed out if and why the improved prevention of nausea is beneficial, even if PM does not inhibit vomiting more efficiently.

Reviewer 2 Report

Many thanks for sending this manuscript to JCM. I have listed my comments on how you should improve your manuscript below:

Materials and methods

  • Instead of "recorded" please write "registered": recorded it at www.clinicaltrials.gov (NCT: NCT01781377).
  • I would appreciate some explanation why it took you 8 years since the study was closed to submit this manuscript?
  • No indication of the dose of local anaesthetic given
  • For how long were the women in trendelebourg position?
  • You gave ephedrine for hypotension treatment; current evidence is favouring either phenylephrine or noradrenaline – please include this in the discussion
  • Please provide a rationale for using NRS for assesing nausea, vomiting and retching – any references?
  • Please provide reference here: "Using data from a previous study of propofol for the prevention of nausea in CS, we 112 calculated that 25 women per group would provide a power of 0.9 and significance level 113 of 0.05 for a reduction of 0.45 in the incidence of nausea."
  • Please include the questionnaire as a supplement/appendix; also the data from the instrument should be in the supplement; what is more, some methodology of the instrument translation/creation needs to be given in the methods section
  • The effect size of Cramers V is reported differently in other sources: "A value for Cramer’s V between 0.06 and 0.17 indicates a small effect, a value between 0.17 and 0.29 is a medium effect, and a value greater than 0.29 is considered a large effect [17]." Here are the references:

Results:

  • Please correct the units for BMI in table 2
  • You fail to acknowledge the big difference in Uterine exteriorization between groups when you describe table 2 results. This should also be discused in the discussion as this is one of the major nausea triggering factors
  • Table 3: you should adjust the significance of P value in post-hoc tests as: 0.05/number of tests; this should also be included in your statistical methos section
  • You mention rescue antiemetics in table 3 – please explain the protocol for the resuce in the methods section
  • Figure 1: please include units on the y axis; why do you use 40 to 65 when your units of measurement were whole integers from 0 to 10?
  • Again, for post-hoc tests you should use adjusted p-values as explained above!
  • Please include the following data in the supplement: "We found no differences in the proportion of patients with a systolic blood pressure 185 below 90 mmHg and a diastolic blood pressure below 50 mmHg (data not shown)."

Discussion:

  • You introduce a new abbreviation IONV here, which was not mentioned before... you should continue with the same names for the main outcomes thruoughout the manuscript. I am not sure the first statement is true once you adjust the p value for multiple comparisons.
  • I would need to re-assess the discsusion once it is improved based on comments in the previous sections of the manuscript
  • Your limitations section should be a bit improved – longer
  • After the limitations section i would add a section on strengths (random allcoation and blinding etc)
  • Your conclusion is a bit contradictory stating that it reduces the proportion of patients with nausea and vomiting in the first sentence, then in the second one you say that the incidence of vomiting was the same

Round 2

Reviewer 2 Report

Dear authors, many thanks for your fast improvement and resubmission of the manuscript. Most of my points have been appropriately addressed, however some issues still remain to be resolved – the major one is concerning multiple comparisons statistics. Please see below - I have added my second round comments to the previous ones and your previous replies. 

Reviewer's comment:

·       No indication of the dose of local anaesthetic given

Authors' response:

Dose is presented in table 2. Please see 152.

Reviewer's comment2:

Please indicate the dosing in or after this sentence too (must be in the methods section too) – what did you base the dose on? Weight-height charts etc… : "We performed SA at levels L 2–3, 3–4 or 4–5 using a 22 G, Whitacre needle and hy- perbaric 5 mg/mL of bupivacaine."

What is more, you should add your method of spinal block height assessment too.

Reviewer's comment:

·       Please provide a rationale for using NRS for assesing nausea, vomiting and retching – any references?

Authors' response:

We found easy to assess it using NRS during cesarean section as women were instructed to use NRS for pain as well. In addition to this, this allowed us to capture the severity of nausea, which is subjective by essence. Please see changes in 100.

Reviewer's comment2: This is sufficient explanation. Also, please add any references too, for example : https://onlinelibrary.wiley.com/doi/10.1111/jocn.14705

Reviewer's comment:

·       Please include the questionnaire as a supplement/appendix; also the data from the instrument should be in the supplement; what is more, some methodology of the instrument translation/creation needs to be given in the methods section

Authors' response:

To our knowledge there is no validated Italian questionnaire for women undergoing CS under SA. doi: 10.1097/ALN.0b013e3182976014. Please see changes in 113. We added the supplement.

Reviewer's comment2: That might be true, but you always have to translate the instrument into italian and then by another native speaker from Italian into English again for the instrument to retain content validity. The original English version must then be compared to the one obtained after double translation. Please add this as a limitation section if you did not use this method.

Reviewer's comment:

Table 3: you should adjust the significance of P value in post-hoc tests as: 0.05/number of tests; this should also be included in your statistical methos section.

Authors' response:

We used the Dunnett’s test over other tests since it affords more statistical power. Drefers to 3 variables, nausea/vomiting/both, 3 post tests (PM vs the other groups), level of significance 0.05. Rows: please see changes in Table 3.

The Dunnett’s test is already adjusted to a level of significance of 0.01 or 0.05. The one tailed test is recommended for known treatments (please see analysis above). However, we decided to be more conservative and presented only two tailed tests.

One side Di results.

Reviewer's comment2: The problem you are having is that you should be only comparing saline/placebo group to P / M  / PM – this is 3 comparisons and not each with eachother which would make 6 comparisons (too small sample size for that and a bunch of other statistical issues). I cite a statistics textbook here: “Dunnett’s test is the only multiple comparison that allows you to test means against a control mean

Anyhow, given your sample size, the control over the type I error is too liberal here. If you chose Dunnett s test, then you should use stricter significance levels: for three comparisons: 0.05/3 = 0.017 (If less than this then it is stat sig). This has to be written in the methods and in all the tables to be clear what is the significance level for multiple testing.

Now that I read table 3, which is your main result, you should have the overall p value – this part is ok, and the three post-hoc tests only: P vs S, M vs S, PM vs S and report the p values of these. Please correct. Moreover, in the text you go on and compare groups PM with P and PM with P (proportions of nausea and vomiting). This is highly selective comparison. The overall comparison with Cramers V is ok, but then you should be comparing the intervention with control and not with another intervention. What is more, the mixed comparisons make the reader confused.

Then for the NRS data: we need some clean results here: what was the mean/median (SD/IQR) NRS score for nausea, vomiting, and NVR? You can do that in the table.

Reviewer's comment:

·       You mention rescue antiemetics in table 3 – please explain the protocol for the rescue in the methods section

Authors' response:

Rescue antiemetics included postoperative ondansetron 4 mg and/or metoclopramide 10 mg. See 85-86.

Reviewer's comment2: Thank you for improving this part. I have a further question : you mention metoclopramide as a rescue antiemetic drug but you have 2 groups which already received metoclopramide (blinded). So I am not sure how good was blinding ? you probably did not repeat the metoclopramide dose.

Reviewer's comment:

·       Please include the following data in the supplement: "We found no differences in the proportion of patients with a systolic blood pressure 185 below 90 mmHg and a diastolic blood pressure below 50 mmHg (data not shown).

Authors' response:

Data included in the supplement.

Reviewer's comment2: Thank you for improving this part. I can not see any supplement other than the Italian version of the questionnaire. Please add the supplements.

Reviewer's comment:

·       Your conclusion is a bit contradictory stating that it reduces the proportion of patients with nausea and vomiting in the first sentence, then in the second one you say that the incidence of vomiting was the same.

Authors' response:

Please see specifications in 282 to 288 and 25

Reviewer's comment2: apologies, but this problem persists in this new version : the second sentence ("The incidence episodes and severity of vomiting was were not affected different among treatment groups.") is contradictory to the first one. It is unclear what you wanted to say. I suggest you delete this second sentence.

Now, concerning discussion and abstract: you should rewrite that to reflect the issues that I pointed out in the results section.

Round 3

Reviewer 2 Report

Dear authors, many thanks for taking into account all the comments. I believe that your manuscript is now much improved and that both types of statistical error are now adequately addressed resulting in appropriate conclusions given the study design and sample size.